

# Short Communication: Evaluating the accuracy of binary classifiers for geomorphic applications

Matthew W. Rossi[1]

[1]Earth Lab, Cooperative Institute for Research in Environmental Sciences, The University of Colorado, Boulder, CO 80303,
USA

*Correspondence to*: Matthew W. Rossi (matthew.rossi@colorado.edu)

**Abstract.** Airborne lidar has revolutionized our ability to map out fine-scale (~1-m) topographic features at watershed- to
landscape-scales. As our 'vision' of land surface has improved, so has our need for more robust quantification of the accuracy
of the geomorphic maps we derive from these data. One broad class of mapping challenges is that of binary classification
where remote sensing data are used to identify the presence or absence of a given feature. Fortunately, there are a large suite
of metrics developed in the data sciences that are well suited to quantifying pixel-level accuracy of binary classifiers. In this
paper, I focus on the challenge of identifying bedrock from lidar topography, though the insights gleaned from this analysis
apply to any task where there is a need to quantify how the number and extent of landforms are expected to vary as a function
of the environmental forcing. Using a suite of synthetic maps, I show how the most widely used pixel-level accuracy metric,
*F1-score*, is particularly poorly suited to quantifying accuracy for this kind of application. Well-known biases to imbalanced
data are exacerbated by methodological strategies that attempt to calibrate and validate classifiers across a range of geomorphic
settings where feature abundances vary. *Matthews Correlation Coefficient* largely removes this bias such that the sensitivity
of accuracy scores to geomorphic setting instead embeds information about the error structure of the classification. To this
end, I examine how the scale of features (e.g., the typical sizes of bedrock outcrops) and the type of error (e.g., random versus
systematic) manifest in pixel-level scores. The normalized version of *Matthews Correlations Coefficient* is relatively
insensitive to feature scale if error is random and if large enough areas are mapped. In contrast, a strong sensitivity to feature
size and shape emerges when classifier error is systematic. My findings highlight the importance of choosing appropriate pixel-
level metrics when evaluating topographic surfaces where feature abundances strongly vary. It is necessary to understand how
pixel-level metrics are expected to perform as a function of scene-level properties before interpreting empirical observations.

## 1 Motivation

The increasing acquisition and access to lidar topography has revolutionized our ability to characterize the fine-scale structure
of the Earth's surface (Roering et al., 2013; Passalacqua et al., 2015). Because lidar can 'see' through the forest canopy, this
technical advance enables quantification of the form and extent of meter-scale features over large areas when mounted on an
airborne platform. Detailed mapping of such features is invaluable to both discovery science and testing hypotheses where the





prevalence of features is expected to vary as a function of the environmental forcing (e.g., in response to changes in climate, ecosystem, rock properties, uplift rates). For example, airborne lidar has now been used to map termite mounds (Levick et al., 2010), mima mounds (Reed & Amundson, 2012), tree throw pits and mounds (Roering et al., 2010; Doane et al., 2021), landslide boundaries and classes (Jaboyedoff et al, 2012; Prakesh et al., 2020), channel network and channel head locations (Pirotti & Tarolli, 2010; Clubb et al., 2014), exposed bedrock (DiBiase et al., 2012; Marshall and Roering, 2014; Milodowski

et al., 2015), and bedrock structure and faulting (Cunningham et al., 2006; Pavlis and Bruhn, 2011; Morell et al., 2017). The utility of lidar topography to these geomorphic applications is unquestioned. These examples also highlight that one of the most common uses for lidar topography is for large-scale, binary classification of finer-scale features. While formal methods for evaluating pixel-level accuracy of binary classifiers is standard practice in the remote sensing and machine learning literature (e.g., Wang et al., 2019; Prakesh et al., 2020; Agren et al, 2021), accuracy assessment in the geomorphic literature is

quite variable. This is likely due to two tendencies of geomorphic studies that employ lidar classifiers: 1. Process-based studies are typically more interested in the *properties* and *densities* of features than their contingent locations; 2. Classifiers are expected to work across *large gradients* in the prevalence of features to test our understanding of the relevant transport laws at play. The former tendency arises from the fact that predicting the actual locations of features (e.g., outcrops, mounds, channels) is not typically a viable target for numerical models of landscapes where uncertainty in initial conditions and the

stochastic nature of processes preclude a deterministic forecasting of surface evolution. The latter tendency arises from the need to use classified data to constrain natural experiments where geomorphic transport laws (Dietrich et al., 2003) can be tested against governing variables (e.g., across climo-, eco-, litho-, or tectono-sequences). I show below that these tendencies can be at odds with pixel-level accuracy metrics that are designed to assess positional accuracy for similarly balanced data (i.e., data where the frequency of positive and negative values does not dramatically vary from case to case).


There are several important benefits to adopting pixel-level metrics when reporting the success of geomorphic classifiers. First, these metrics provide common standards for evaluating classifier accuracy across studies, including direct comparison between proxy-based classifiers with those developed using machine learning. Second, trends in pixel-level accuracy scores may reveal distinct patterns in the spatial structure of error. Third, pixel-level measures are easy to apply to new objectives as long their

limitations are properly considered. To this end, I focus on how two widely used metrics, *F-measures* (van Rijsbergen, 1979; Chinchor, 1992) and *Matthews Correlation Coefficient* (Matthews, 1975; Baldi et al., 2000), perform when the research design intentionally calibrates and tests binary classifiers across large gradients in how balanced the data are. Interactions among feature size, feature shape, and error structure can produce diagnostic trends in accuracy scores as a function of feature prevalence. As such, I argue here that pixel-level accuracy scores should be evaluated alongside performance at other scales,

particularly the scene-level scale where the statistical attributes of features can be quantified for a given environmental forcing.





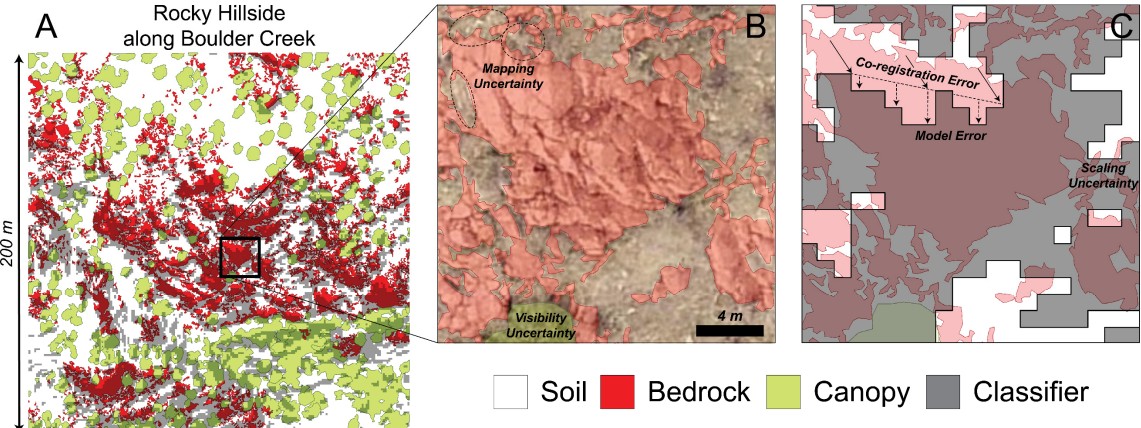

**Figure 1:** Example bedrock mapping from Rossi et al. (2020) showing (A) a classified scene. Zoom boxes illustrate different kinds of error
due to (B) mapping 'truth' from air photos and (C) using coarser resolution lidar data to 'model' bedrock. In A, scene-level patterns in actual
bedrock exposure were mapped using 3-inch Pictometry® air photos and a topographic proxy derived from 1-m airborne lidar (Anderson et
al., 2012) as the classifier. The bedrock fraction mapped from air photos is 0.24. The bedrock fraction mapped from lidar data using a
regionally based, slope-threshold of 38° is 0.35. The zoom area used for B and C is shown in A as a black box. In B, the truth map for this
site is overlaid on associated air photos at 75% transparency to show the two principal sources of error in air photo mapping. In C, the
bedrock classifier is overlaid on the same truth map to show the three principal sources of error in using the lidar data for classification.

## 2 Example application: Bedrock mapping

The task of mapping bedrock outcrops is useful to show how pixel-level accuracy metrics can be applied to geomorphic studies
for a few reasons. First, the transition from fully soil-mantled to bedrock-dominated hillsides reflects an important continuum
in process dominance and rates (Heimsath et al., 2012). Whether and where bedrock is observed records the local (im)balance
between soil production and denudation rates (Gilbert, 1909), providing important tests to hypothesized soil production
functions (e.g., exponential versus 'humped'; Heimsath et al., 1997; Anderson, 2002) and sediment transport laws (e.g., linear
versus nonlinear creep; Culling, 1963, Andrews & Bucknam, 1987). Second, the challenge of mapping bedrock using airborne
lidar data is an application that has received a fair bit of recent attention (DiBiase et al., 2012; Marshall & Roering, 2014;
Milodowski et al., 2015; Rossi et al., 2020). This is, in part, because individual bedrock features can be resolved in lidar
topography using physically interpretable slope and roughness thresholds. Airborne lidar balances trade-offs between data
resolution (~1-m) and data coverage (100's of km$^2$) and thus is well-suited to exploring how feature density and properties
vary across environmental gradients. Third, identifying bedrock typifies the more general challenge of understanding the
related, but distinct, scaling properties among data, features, and processes (Sofia, 2020). Bedrock tors and cliffs occur at many
scales (sub-meter to tens of meters) that reside on hillsides (hundreds of meters in length) which are, in turn, responding to
base level signals propagating through river networks (tens to thousands of km$^2$).



In this paper, I specifically consider the approach taken by DiBiase et al. (2012) and adopted by Rossi et al. (2020). These studies calibrated lidar proxies for bedrock in the San Gabriel Mountains, CA, USA (SGM) and the Colorado Front Range, CO, USA (CFR), respectively. The general approach in both was to map bedrock using photographic imagery for 50 x 50 m to 200 x 200 m patches where the ground surface is visible due to limited forest cover and/or recent clearing due to wildfire. By selecting scenes representative of a large range of bedrock fractions, the main goal of these studies was to identify a single slope threshold that could be applied across the landscape. Both studies found strongest correlations using slope-thresholds somewhat above the angle of repose for granular materials (45° in the SGM and 43° in CFR). However, the threshold that most closely reproduced the scene-level bedrock fraction without rescaling is closer to expected values for the angle of repose (e.g., a slope-threshold of 38° produced a regression slope of one in the CFR; Rossi et al., 2020). Regressions in the CFR were overall weaker, likely due to differences in air photo mapping (1-10 cm surface-normal field photos in the SGM versus ~8 cm air photos in the CFR) and the increased prevalence of bedrock tors, or isolated bedrock outcrops within a lower relief soil mantle. Bedrock tors tend to produce dome-shaped features with steep slopes on their sides and low-sloped tops that may be better resolved using roughness-based topographic proxies (Milodowski et al., 2015). While scene-level success of slope-based proxies for bedrock in the SGM (peak $r^2$ of 0.99) and CFR (peak $r^2$ of 0.85) are promising, neither study assessed pixel-level accuracy.

Figure 1 shows two general challenges associated with using air photos to calibrate and validate lidar-based proxies for bedrock exposure. The first general source of error is introduced in the generation of 'truth' data from air photos (Fig. 1B). Even under the best circumstances, **visibility** of the ground surface is obstructed in places by the vegetation canopy. This can be partially addressed by restricting mapping tasks to areas where obstructions are minimal and ground truthing air photo mapping with field observations. While using high resolution air photos aids interpretation, it is difficult to fully eliminate human error in **mapping** due to shadows or weakly contrasting visible properties between bedrock and soil. Similarly, distinguishing in-place bedrock from detached coarse sediment is difficult unless coarse sediment collects into macro-scale features, like talus slopes, whose properties are distinct. The second general source of error arises in the classification process itself (Fig. 1C). Relating higher resolution air photos to lidar proxies requires better understanding of uncertainty in the **scaling** properties of features. Scaling challenges arise from both the feature shape itself and how gridded representations of features change as a function of data resolution. Because the classifier is often built from data acquired at different times and using different data sources, error in classification can also arise due to **co-registration** of truth and model datasets. Precise mapping of control points for georeferencing and smart use of stable surfaces in post-processing can help minimize the misfit between truth and model data (Bertin et al, 2022). The binary classifier itself, whether using physical thresholds or statistical models, will also be imperfect. New algorithms attempt to make this **model** error as small as possible. Each of these five sources of error lies on a continuum between random and systematic, where random error is independent of feature locations or properties and systematic error refers to any error structure that is spatially correlated with feature locations or properties. For example, we might expect co-registration error between two remote sensing datasets to be more systematic than the others due to translation, rotation, and



distortion of aligned datasets. Can pixel-level accuracy scores diagnose different error structures when calibration of binary classifiers is attempted against scenes that span large gradients in bedrock exposure? How does feature shape, feature scale, and mapping coverage interact with this error structure?

## 3 Approach

Two of the most widely used accuracy metrics are *F1-score* and *Matthews Correlation Coefficient* (*MCC*). Adopting such pixel-level metrics helps link studies that classify features using physical intuition (e.g., using slope-thresholds for bedrock exposure is based on the notion that only bedrock is stable above the angle of repose) with those developed using statistical methods (e.g., machine learning). These measures also provide a common language to assess results from studies that span different landscapes with different research goals. However, I emphasize here that while these metrics can robustly characterize

pixel-level accuracy, it is important to consider their limitations in characterizing scene-level accuracy and how they might perform across gradients in environmental forcing. To this end, I consider a suite of synthetic land surfaces that show the sensitivity of *F1-score* and a normalized version of *MCC* to: feature scale, the error structure in the data, and how balanced the data are. In Section 3, I describe methods common to all scenarios. Specifically, I describe the general process of generating grids and calculating accuracy scores. Methods unique to each different scenario are then described in Sections 4 and 5 so that

their rationale can be articulated in the context of results.

### 3.1 Grid generation

To generate 'truth' grids of bedrock and soil, I first use the pseudo-random number generator in NumPy to create a scene of size $m$ x $n$ cells. Continuous values are converted into binary classes (0 = soil; 1 = bedrock) based on a user-specified value for the overall fraction of bedrock ($f_b$). The simplest scenario is for bedrock tors with a size of one pixel. While synthetic

surfaces are scale free, I report results assuming a grid spacing of 1-m to represent a typical case using airborne lidar. To simulate features that have a scale greater than one square meter, I use the pseudo-random numbers to instead specify a first guess at the locations of the centres of incipient tors. The first guess at the number of tors is calculated by finding the integer number of tors of length, $l$, that most closely matches $f_b$. However, as the number of tor centres increases, so does the probability that two neighbouring 'tors' overlap and coalesce into a larger feature. As such, the first guess generally produces

an actual bedrock fraction lower than the user-specified value. The ratio between the specified $f_b$ and this underestimate is then used to proportionally increase the number of incipient tors in the model domain. I iterate this process until either the synthetic fraction is within 0.5% of the specified value or fifty iterations, whichever comes first. It is worth pointing out here that while I will continue to use the term 'tor density' to refer to the number of tor centres per scene area, the resultant number of tors is smaller due to the coalescing of incipient features (see section 6.2 and Appendix B2 for further elaboration).






All scenarios presented in this study rely on comparing simulated 'truth' and 'model' grids. Where the truth and model data are independent of each other, the two grids are generated using different pseudo-random seed numbers in NumPy (section 4). In scenarios where the model grid is dependent on the truth grid, the model grid is a copy of the truth data using the specified error structure. Details for how random error (section 5.1), systematic error (section 5.2), and random plus systematic error

(section 5.3) are implemented are described in context below. For each scenario, the truth and model grids are evaluated by building the confusion matrix and calculating accuracy metrics at each bedrock fraction (section 3.2).

### 3.2 Pixel-level accuracy metrics

While there are many metrics used to quantify the accuracy of binary classifiers, I focus here on two widely used ones: the *F1-score* and *Matthews Correlation Coefficient* (*MCC*). These metrics can be used to evaluate pixel-level performance of

classified maps with respect to ground truth data and are often used when employing machine learning techniques (Wang et al., 2019; Prakesh et al., 2020; Agren et al, 2021). Application of these metrics need not be limited to the training and testing of machine learning algorithms. They are broadly useful to any binary classification task where positional accuracy is important. Both *F1-score* and *MCC* can be calculated directly from the confusion matrix. The confusion matrix for binary classification is a 2x2 table where the column headers are the true classes and the row headers are the model classes, thereby

summarizing the occurrence of the four possible classification outcomes: True Negatives (TN), True Positives (TP), False Positives (FP), and False Negatives (FN). For example, the scene in Figure 1A can be readily reclassified into these four outcomes (Fig. 2A) which is summarized by the confusion matrix shown in the inset of Figure 2B. The simplest assessment of accuracy is the overall accuracy (OA), and its complement the error rate (ER), where:

$$OA = \frac{TP+TN}{TP+TN+FP+FN} \tag{1}$$


$$ER = \frac{FP+FN}{TP+TN+FP+FN} \tag{2}$$

While *OA* and *ER* are straightforward to calculate, they provide little insight into the relative frequencies of FP and FN. To address this limitation, there are a large family of accuracy metrics that better characterize different types of error. For example, *precision* and *recall* characterize the relative frequencies of FP and FN explicitly. *Precision*, also known as the positive predictive value, is the ratio of true positives to all positives predicted by the model (accounts for FP) and *recall*, also known

as the true positive rate, is the ratio of true positives to all positives (accounts for FN), whereby:

$$Precision = \frac{TP}{TP+FP} \tag{3}$$

$$Recall = \frac{TP}{TP+FN} \tag{4}$$

Figure 2 is an example where the *precision* is low (0.36), but the *recall* is reasonably good (0.58) (Table 1). *F-measures* were designed to summarize *precision* and *recall* into a single metric (van Rijsbergen, 1979; Chinchor, 1992). The case where both

are equally weighted is referred to as the *F1-score*, where:

$$F1\text{-}score = \frac{2\times TP}{(2\times TP)+FP+FN} \tag{5}$$



By representing the harmonic mean of *precision* and *recall*, this metric accounts for both errors of omission and commission. However, *F1-scores* only characterize the success at identifying the target class, and low values can occur even if the overall accuracy is high because it excludes True Negatives. Consequently, this metric is quite sensitive to the prevalence of positive

values whereby higher scores are favoured when the positive class is more abundant (e.g., Chicco and Jurman, 2020). Related to this sensitivity to imbalanced data is the property of asymmetry. Asymmetric metrics are those where the accuracy score differs when the target classes are switched. Table 1 shows that the *F1-score* for Figure 2 would be 72% higher if the target class was soil instead of bedrock. Asymmetry arises because there is more soil than bedrock in the scene and TN are not included in calculations of *precision*, *recall*, or *F1-score*. These well-known limitations of *F-measures* are better handled by

metrics that incorporate all four classes of the confusion matrix. One such metric is *Matthews Correlation Coefficient* (*MCC*), where:

$$MCC = \frac{(TP \times TN) - (FP \times FN)}{\sqrt{(TP+FP) \times (TP+FN) \times (TN+FP) \times (TN+FN)}} \tag{6}$$

*MCC* is equivalent to a Pearson's correlation coefficient where the model classes are regressed against the true classes in a binary classification task (Fig. 2B). Values of *MCC* can be similarly interpreted where -1.0 indicates perfect anti-correlation,

0 is a random model, and 1.0 indicates perfect correlation. And while *MCC* is just one of several metrics that include all four quadrants of the confusion matrix (e.g., Balanced Accuracy, Markedness, Cohen's Kappa), *MCC* appears to be the most robust to imbalanced data (Chicco and Jurman, 2020; Chicco et al., 2021a; Chicco et al., 2021b). In this analysis, I report the normalized version of *MCC* as:

$$nMCC = \frac{MCC+1}{2} \tag{7}$$

By re-scaling *MCC* from zero to one, *nMCC* facilitates comparison with *F1-score* on plots and in discussion. It is worth noting here though that interpretations of low values of *nMCC* differ from interpretations of low values of *F1-score*. The former implies anti-correlation between model and truth data while the latter does not. For example, the scene in Figure 2 indicates a weak positive correlation (i.e., *nMCC* greater than 0.5) even though the *F1-score* is lower than 0.5 (Table 1). As such, direct comparison of these metrics should be done with caution.


**Table 1:** Accuracy metrics for Figure 2 using the alternative target classes of bedrock and soil.

| Target Class | *OA** | *ER** | *Precision* | *Recall* | *F1-score* | *MCC** | *nMCC** |
|---|---|---|---|---|---|---|---|
| Bedrock | 0.67 | 0.33 | 0.36 | 0.58 | 0.44 | 0.24 | 0.62 |
| Soil | 0.67 | 0.33 | 0.85 | 0.69 | 0.76 | 0.24 | 0.62 |

\* *Metrics that do not vary as a function of the target class in binary classification.*



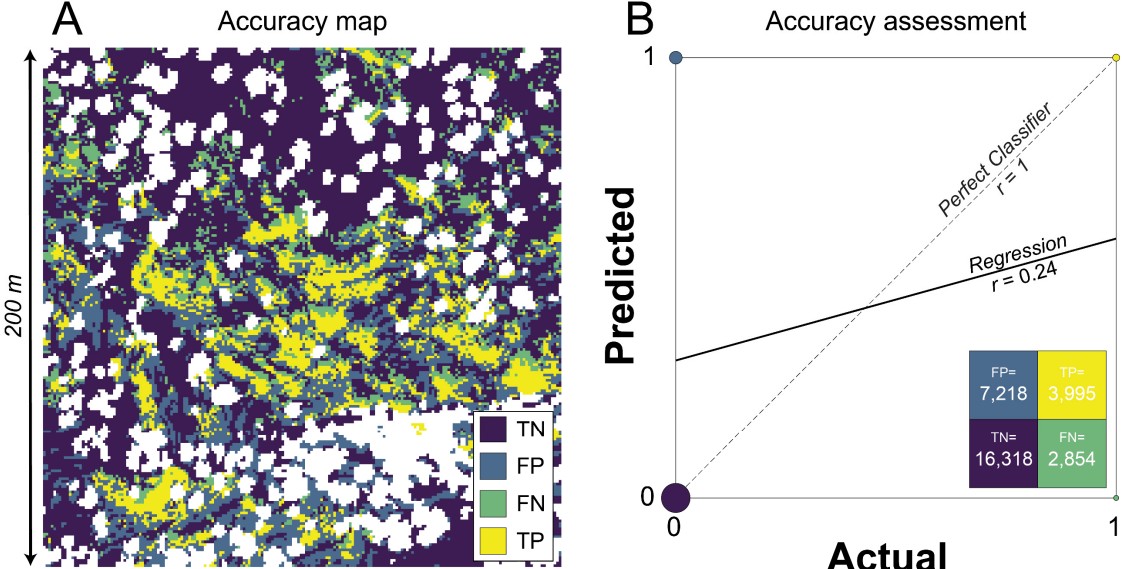

**Figure 2:** (A) Pixel classes for Fig. 1A and (B) the corresponding confusion matrix (inset) and correlation plot (main). In A, the four outcomes of the binary classification are shown in colour [TN = True Negatives; FP = False Positives; FN = False Negatives; TP = True Positives]. The areas in white were obscured by the vegetation canopy in air photos (24% of area) and thus excluded from accuracy assessment. In B, the colours of each cell in the confusion matrix and each point in the plot are the same as in A. The number of observations for each class is shown in the confusion matrix and point sizes on the plot are scaled to the relative frequency of each value.

## 4 Pixel-level versus scene-level accuracy

Throughout this analysis, I distinguish between pixel-level and scene-level measures of accuracy. Pixel-level accuracy requires that the precise locations of features are honoured where the lower bound to feature detection is set by the spatial resolution of the data used. Scene-level accuracy characterizes the mismatch between model and truth data at some coarser scale and typically assesses statistical properties of the target feature class (e.g., bedrock fraction, mound densities, drainage densities).

While high pixel-level accuracy ensures high scene-level accuracy, the converse need not be true. This distinction is motivated by studies where calibration of lidar classifiers was undertaken only at the scene-level (DiBiase et al., 2012; Rossi et al., 2020) and whose rationale was summarized in section 2. Scene-level assessment alone may lead to different findings than pixel-level assessment. For example, a related effort by Milodowksi et al. (2015) showed that lidar-based, roughness-thresholds provide an alternative topographic proxy that can be more successful than slope-based ones in some landscapes. While their overall objective of finding a classifier that worked across a range of bedrock fractions was similar to DiBiase et al. (2012), these authors used pixel-level assessment to select thresholds and evaluate classifier success. As such, there is a need to understand how pixel-level assessments behave when classifying data where scenes are intentionally selected across gradients in how balanced the data are.



Figure 3 shows the sensitivity of *F1-score* and *nMCC* to a research design that tests binary classifiers across gradients in bedrock fraction for a typical scene ($m = n = 100$). Two scenarios are considered. Each assume bedrock outcrops are randomly distributed throughout the scene for any given bedrock fraction. In the first scenario, the bedrock classifier predicts that bedrock is found everywhere regardless of the truth data (dashed lines). Because this 'all rock' model produces neither False Negatives nor True Negatives, *nMCC* is undefined in this scenario (see eqs. 6-7). *F1-score* nonlinearly improves with increasing bedrock

fraction and approaches unity as the actual bedrock fraction nears the 'all-rock' model. In the second scenario, the bedrock classifier is forced to match the bedrock fraction in the truth grid, though the locations of bedrock outcrops in the model are independent from the truth data (solid lines). This represents a worst-case scenario for a classifier that successfully models the scene-level fraction of bedrock while also providing zero predictive value at the pixel level. The values of *nMCC* rightly diagnose independence between the model and truth data by showing zero correlation across the full range of bedrock fractions

(*nMCC* ~ 0.5). *F1-score* increases as a linear function of bedrock fraction. As this and subsequent examples show, *F1-score* embeds a spurious correlation with bedrock fraction, all other things being equal, because the number of True Negatives is ignored. In contrast, *nMCC* provides a robust metric to evaluate positional error for classifiers that have been calibrated to scene-level properties like bedrock fraction. While these relationships do not depend on tor size, larger mapping areas are needed to adequately sample the statistics of feature locations when incipient tors are large with respect to the area of the scene

(Fig. 3D). The noisy relationships in Figure 3D largely reflect the inability to match the specified bedrock fraction using a discrete number of random tors whose locations are set by the specific pseudo-random seed used. In fact, 49% of the grids generated for Figure 3D failed to meet the 0.5% tolerance of specified bedrock fractions after fifty iterations. For subsequent analyses, I use larger scenes of 1000 x 1000 m to mitigate the effect of domain size on accuracy scores. For this larger domain, nearly all (>99%) the subsequent grid pairs meet the tolerance criterion before fifty iterations, which manifest as smoother

curves in plots.





**Figure 3:** Classified 100 x 100 m maps of (A) 1-m and (B) 10-m long square tors showing the four classification outcomes (*TN*: True Negatives, *FN*: False Negatives, *FP*: False Positives, *TP*: True Positives). How accuracy scores vary as a function of bedrock fraction are also shown for (C) 1-m and (D) 10-m long tors, respectively. The 'all rock' scenario is where the model data assumes the entire surface is bedrock regardless of the actual bedrock fraction. The 'match scene' scenario is where the model data matches the actual bedrock fraction, but whose locations are independent. In A-B, example maps are shown for the case of 0.5 bedrock. In C-D, normalized Matthews Correlation Coefficient (*nMCC*) is only shown for the 'match scene' scenario because it is undefined in the 'all rock' scenario.



**5 Error structure**

In the previous section, I showed how *F1-score* and *nMCC* vary as a function of bedrock fraction for very poor pixel-level

classifiers. A good classifier though, whether statistically or physically based, should be successful in most cases with some residual error. To illustrate these more realistic conditions, I consider three scenarios where the error structure is either random (section 5.1), systematic (section 5.2), or both (section 5.3). While actual sources of error are complex (e.g., Fig. 1), these endmember scenarios are intended to provide a heuristic understanding for how pixel-level accuracy scores perform when the research design explicitly samples across a gradient in feature prevalence.

**5.1 Random error**

The first error scenario I consider is the situation where the binary classifier successfully identifies bedrock with a fixed rate of random error ($\bar{e}_r$). To create synthetic surfaces of this type, a truth grid is first generated (for Fig. 4 $m = n = 1,000$) for a given bedrock fraction. Bedrock tors are assumed to occupy a single pixel, though results are robust to different sizes of incipient tors because the error location is independent of feature location. To produce the associated model grid, I first

generated an error grid using a different pseudo-random seed than that used to generate the bedrock grid. The grid of continuous values is converted to binary classes (0 = no error; 1 = error) using the specified error rate as the threshold. The error grid is then used to construct the model grid from the truth grid by flipping bedrock classifications wherever the error grid value equals one. Note that the maximum error rate shown in Figure 4 is 0.5. This is the scenario where the truth and model data are least correlated. Increasing the error rate further will produce increasingly stronger negative correlations between model and

truth data. Once both truth and model grids are generated, *F1-score* and *nMCC* are calculated. This analysis is done for bedrock fractions that range from 0.01 to 0.99 and error rates from 0.05 to 0.5.

I show the results of this analysis for ten different error rates in Figure 4. These results can be derived analytically from eqs. 5-7 and the imposed random error rate (Appendix A). I use results from synthetic landscapes: 1. To ensure that synthetic scenes

adequately sample population statistics; and 2. Facilitate integration with scenarios that include non-random error (section 5.3). As should be expected, Figure 4 shows that accuracy scores increase with lower error rates. However, the sensitivity of these scores is not uniform with respect to bedrock fraction. Much like in the previous example (Fig. 3), *F1-scores* always monotonically improve with increasing bedrock fraction. Note here though that the worst case (Fig. 4 dashed black line; 50% error rate) is not equivalent to the case where the model is independent from the truth data (i.e., the solid black line in Fig. 3).

In the random error scenario, model data are correlated with, but not equal to, actual bedrock fractions (Fig. 4A). The fixed error rate preferentially modifies the larger frequency class when near the endmember cases of all bedrock and all soil. This behaviour is easily envisioned for the case where the error rate is 50%. If the actual surface is all bedrock, then the model produces 50% soil on average, and visa versa if the actual surface is all soil. In the random error scenario, the slope of the relationship between modelled and actual bedrock equals $1 - 2\bar{e}_r$ (Appendix A). The symmetry of the sensitivity of *nMCC* to



a constant error rate allows for comparison of map accuracies across a wide range of differentially balanced data, specifically

over the domain over which *nMCC* is approximately invariant (Fig. 4B). In contrast, disentangling the spurious correlation

between *F1-score* and bedrock fraction interacts with the preferential modification of surface classes in a complex way, leading

to increasing nonlinearity in response to better classifiers with lower error rates.

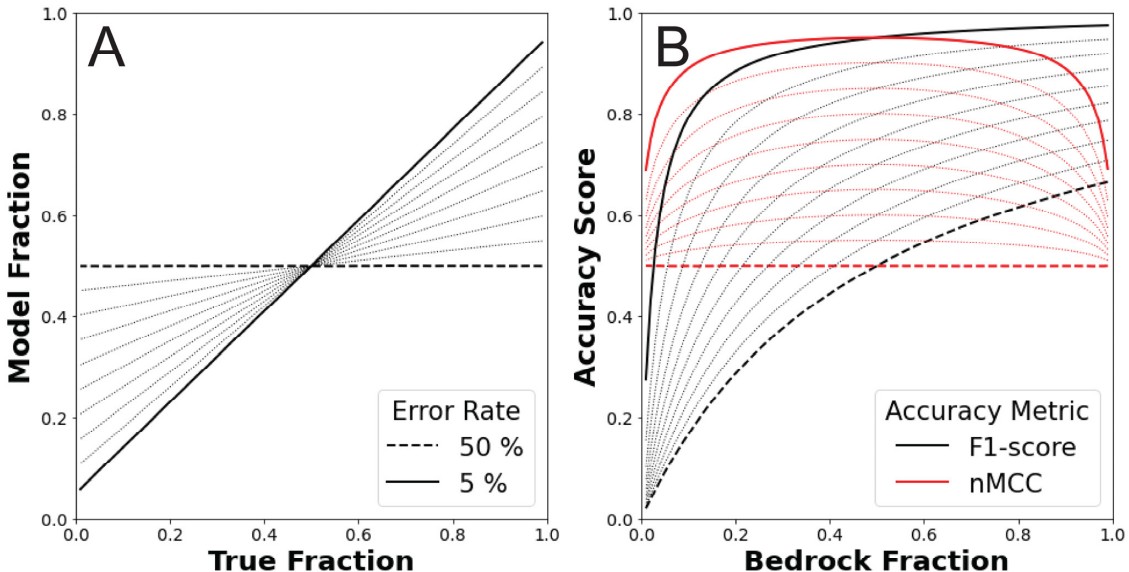

**Figure 4:** (A) Model bedrock fractions and associated (B) accuracy scores as a function of true bedrock fraction for the random error scenario
(1000 x 1000-m map area). In both plots, the minimum and maximum error rates are highlighted, and 5% increments of error rate are shown
as dotted lines. In A, matching the model fraction to the actual fraction of bedrock is not enforced like in other scenarios (Figs. 3, 5).
However, the two fractions are linearly related, and the slope of the relationship is directly related to the error rate. In B, lower error rates
amplify the nonlinearity between *F1-score* and bedrock fraction while *nMCC* more uniformly improves across a broad range of bedrock
fractions.

**5.2 Systematic error**

The second error scenario I consider is the situation where the binary classifier successfully identifies bedrock with systematic

error. To create synthetic surfaces of this type, a truth grid is first generated (for Fig. 5 $m = n = 1,000$) for a given bedrock

fraction and tor size. Incipient tors are randomly distributed throughout the model domain. To generate the associated model

grid, a copy of the truth grid is linearly offset by one pixel to the right in the x-direction, though results are insensitive to the

direction of the shift. By using wrap-around boundaries, synthetic truth and model grids always have an identical bedrock

fraction. Note that the systematic error rate ($\bar{e}_s$) is not constant and is a function of the bedrock fraction, the magnitude of the

systematic offset, and the shape and size of features. Once both truth and model grids are generated, *F1-score* and *nMCC* are

calculated. This analysis is done for bedrock fractions that range from 0.01 to 0.99 and for tor sizes that range from 1x1 to

10x10 (i.e., areas of 1 to 100 pixels).



I show the results of this analysis for ten different tor sizes in Figure 5. While I discuss results in terms of a scale typical to airborne lidar (i.e., 1-m spatial resolution), the relationships shown here are better cast as the ratio of the feature scale (tor length) to the error scale (1 pixel length) where the feature detection limit is one pixel. When the error is of order feature

length, systematic error mimics the case where the truth and model data are independent (e.g., compare long dashed lines in Fig. 5 to solid lines in Fig. 3). As the systematic error gets smaller with respect to the tor size, both *F1-score* and *nMCC* improve. The largest improvements occur for small tor sizes and at low bedrock fractions (Fig. 5B). When bedrock fractions are low, the error is largely due to the geometric effect of the shift of individual square tors surrounded by soil such that $TP = \frac{l^2-l}{l^2}$ and $FP = FN = \frac{l}{l^2}$ (Appendix B). As bedrock fraction increases, incipient tors increasingly coalesce into a smaller number

of features, and the error is set by these more complex geometries (see discussion in section 6.2). Figure 5A shows how increasing tor sizes leads to lower error rates and increasing asymmetry in error as a function of bedrock fraction. Error rate functions skew towards higher bedrock fractions leading to a modest negative relationship between *nMCC* and bedrock fraction (Fig. 5B). The asymmetric error structure also impacts *F1-score*, albeit in a way that is much harder to diagnose due to the spurious correlation between *F1-score* and bedrock fraction (Figs. 3-4). The notion of systematic error in scene-level mapping

was envisioned for situations where co-registration error between the remote sensing data used to map 'truth' and the remote sensing data used to build the classifier produce a systematic, translational offset. Strictly speaking then, this synthetic scenario represents the case where a translational offset is the same for all scene-level patches, a plausible situation if the truth and model data for different scenes were acquired at the same time and in the same way. However, even under the less stringent condition where co-registration errors are oriented differently in different scenes (i.e., due to different acquisition parameters

and times), the relationships shown in Figure 5 will still hold as long as the magnitude of the systematic error is similar across sites and the orientation of features is isotropic.



Earth **Surface**
**Dynamics**
Discussions

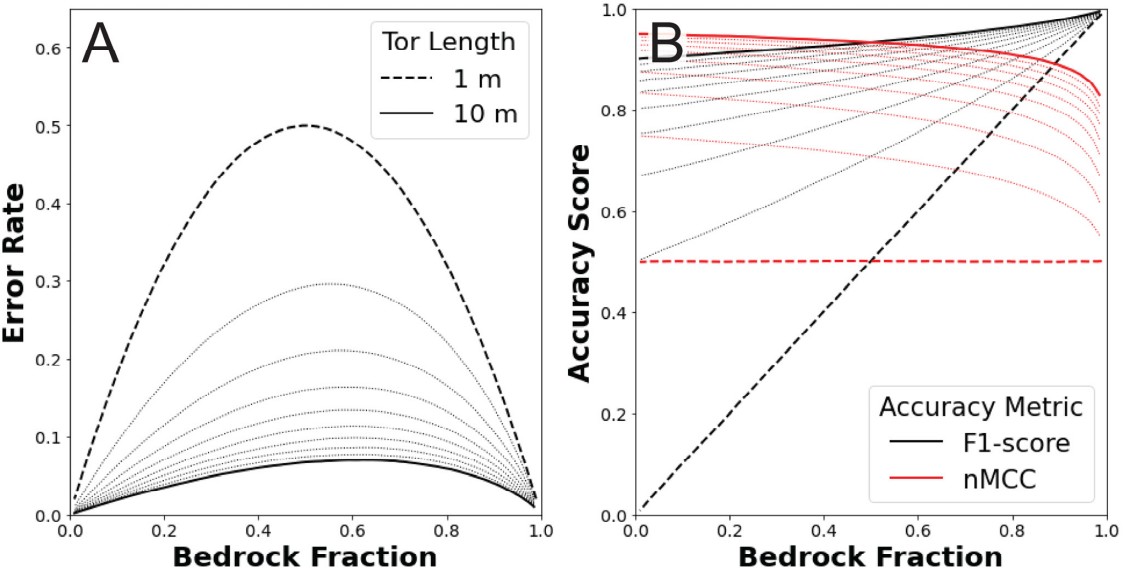

**Figure 5:** (A) Error rate response to tor size and associated (B) accuracy scores as a function of the true bedrock fraction for the systematic
error scenario (1000 x 1000 m map areas). In both plots, the minimum and maximum tor lengths are highlighted, and 1-m increments are
shown as dotted lines. In A, the error rate (eq. 2) is non-uniform with lower rates at both low and high bedrock fractions. As tors get larger,
the error rate function becomes increasingly asymmetrical with peak values at 0.5 and 0.66 bedrock for 1- and 10-m long tors, respectively.
In B, the non-uniform error rates lead to more linear relationships between *F1-score* and bedrock fraction than in the case of random error
(Fig. 4B). In contrast, *nMCC* shows modest negative relationships with bedrock fraction for all incipient tor sizes.

### 5.3 Random plus systematic error

The third error scenario I consider is the situation where the binary classifier is systematically offset from the truth grid with
an additional random error term. To create synthetic surfaces of this type, a truth grid is first generated (for Fig. 6 m = n =
1,000) for a given bedrock fraction ($f_b$) and tor size. Tors are randomly distributed throughout the model domain. To generate
the associated model grid, a copy of the truth grid is first linearly offset by one pixel to the right in the x-direction, using a
wrap-around boundary condition. I then generate a random error grid using a different pseudo-random seed than that used to
generate the bedrock grid. The grid of continuous values is converted to binary classes (0 = no error; 1 = error) using a random
error rate of 0.05 as a threshold. The random error grid is then used to flip bedrock classifications in the offset grid wherever
the error grid value equals one. Note that model bedrock fractions need not match the truth data, and error rates will now be a
function of the bedrock fraction, the magnitude of the systematic offset, the shape and size of bedrock features, and the random
error rate. Once both truth and model grids are generated, *F1-score* and *nMCC* are calculated. This analysis is done for bedrock
fractions that range from 0.01 to 0.99 and tor sizes that range from 1x1 to 10x10 (e.g., areas of 1 to 100 pixels).



Figure 6 is analogous to Figure 5 with error rates (Fig. 6A) and accuracy scores (Fig. 6B) plotted as a function of bedrock

fraction for different tor sizes. The random error rate sets the minimum observed error and contributes to the total error in a

nonlinear way. This is because the random error term can flip values where systematic error occurs (i.e., both sources of error

can combine to produce True Positives). Figure 6C and 6D show the differences in error rates and accuracy scores, respectively,

between systematic error alone (Fig. 5A-B) and the systematic error plus random error scenario (Fig. 6A-B). The addition of

random error is more influential in cases where the classifier is more accurate (i.e., larger tors) and near endmember bedrock

fractions (i.e., all soil and all rock). For a given incipient tor size, the minimum error added by the random error rate of 0.05

occurs at intermediate bedrock fractions and ranges from near zero for 1-m tors to 0.043 for 10-m tors. The results shown in

Figure 6B show that relationships between pixel-level accuracy scores and scene-level bedrock fraction for this scenario

include elements of both the random error and systematic error scenarios. Because random error is the dominant term of the

total error rate near the endmember cases of zero and all bedrock, it leads to correspondingly large reductions in $nMCC$ (Fig.

6D). In contrast, at intermediate bedrock fractions there is slight negative slope to $nMCC$ like observed in the systematic error

scenario (Fig. 5B). This is because reductions in $nMCC$ induced by random error at intermediate bedrock fractions are:

relatively smaller, approximately invariant across a broad range of fractions, and symmetrical with respect to bedrock fraction

(Fig. 6D). While I only show one random error rate, this example shows how complex interactions between random and

systematic error can be readily simulated. While $nMCC$ is strongly preferred over $F1\text{-}score$ when building and testing

classifiers across gradients in feature abundance, the two scenarios that include systematic error suggest that asymmetry in

accuracy scores can still arise in response to the geometries and genesis of more features. In these cases, asymmetry is not due

to limitations of the accuracy metric, but instead a result of how features are simulated in synthetic examples. Whether my

synthetic generative process (i.e., randomly distributed square tors of constant size) is representative of real transitions from

soil-mantled to bedrock-dominated hillsides is an open question. However, these synthetic examples provide an opportunity

to probe how the evolution of feature geometries influence accuracy scores, a topic that is explored in much more depth below

and in Appendix B.

Earth **Surface**
Dynamics
Discussions



**Figure 6:** (A) Error rate response to tor size and associated (B) accuracy scores as a function of the bedrock fraction for the systematic plus random error scenario (1000 x 1000-m map areas). These panels are analogous to Figure 5A and 5B but now include a 5% random error
term. Differences in (C) error rate and (D) accuracy scores between this figure and Figure 5 are shown to enable comparison. In C, the additional 5% random error term is linearly added to the systematic error term at the end-member cases of zero and all bedrock. The random error translates into something less than the 5% additional error at intermediate cases with minima near zero for 1-m tors and 0.043 for 10-m tors. In D, *nMCC* exhibits strong reductions from systematic error alone near endmember cases (high negative values) and a muted, more uniform reduction for intermediate values.




## 6 Discussion

### 6.1 Accuracy assessment for imbalanced mapping tasks

Mapping patchy bedrock exposure is a good use case for binary classification on imbalanced data. Many studies have now had success doing scene-level mapping of bedrock exposure using lidar topography (DiBiase et al., 2012; Heimsath et al., 2012; Marshall et al., 2014; Milodwoski et al., 2015; Rossi et al., 2020). By calibrating lidar classifiers at the hillslope scale, there are enough observations to characterize the statistics and properties of bedrock features while also minimizing intra-scene variations in climate, ecosystem, rock properties, and base level controls on soil production and denudation rates. Of this prior work, the only one to use pixel-level accuracy scores to calibrate and validate their bedrock classifier was Milodowski et al. (2015). In their analysis, lidar classifiers were assessed at multiple roughness thresholds applied over different spatial neighbourhoods. Recognizing the challenges of imbalanced data, these authors subsampled the more frequent class in each scene to match the number of observations of the smaller class. My analysis shows that *Matthews Correlation Coefficient* (*MCC*) provides an alternative approach to handling the challenge of comparing scenes with different bedrock fractions that also addresses the problem of asymmetry embedded in other pixel-level metrics (Table 1). While the limitations of metrics like *F1-score* are already well-known (Chicco and Jurman, 2020), Figure 3 emphasizes an important implication of using this metric when the research design intentionally samples across scenes with differentially balanced data. Adding scene-level constraints to a random classifier leads to lower *F1-scores* than simply assuming the entire surface is bedrock. In other words, adding scene-level information in the calibration process actually reduces *F1-score*. This vulnerability is true for all pixel-level assessments that do not consider all four components of the confusion matrix (e.g., *precision*, *recall*, *F-measures*, *receiver operating characteristic curves*).

The results presented here corroborate arguments that *MCC* is generally a more robust pixel-level accuracy metric than *F1-score* (Chicco & Jurman, 2020), specifically within the context of calibrating and validating bedrock mapping algorithms. Despite the improvements afforded by *MCC*, caution is still warranted in directly interpreting how pixel-level metrics will vary as a function of feature prevalence. Uniform, random error preferentially modifies the dominant class, leading to strong reductions in accuracy near endmember cases, all other things being equal (Fig. 4; Appendix B1). Even for accurate classifiers, random error limits the domain over which *MCC,* and thus *nMCC*, can be confidently compared at the scene-level (e.g., accuracy scores for 5% random error stabilize between ~20 to 80% bedrock; Fig. 4). Furthermore, linear regressions of observed and classified bedrock fractions (e.g., DiBiase et al., 2012; Rossi et al., 2020) can provide clues as to how error varies across scenes. Under the narrow conditions of uniform and spatially random error, the y-intercepts of regressions should equal the error rate and the regression slope should be less than one (Appendix A). The linear regressions presented in Rossi et al. (2020) were forced through the origin. Had they not been, the y-intercepts of those fits would have been negative, suggesting that the classified lidar data tended to produce more error at lower bedrock fractions. While the number of scenes analysed was small (8 scenes), this tendency towards higher error at lower bedrock fractions makes sense with respect to how bedrock





emerges in the Colorado Front Range. Lower relief hillsides with less bedrock are dominated by tors as opposed to bands of

bedrock cliffs that begin to emerge on higher relief hillsides. The myriad sources of error in real landscapes (Fig. 1) will lead

to much more complex intra-scene error than either the random or systematic error scenarios posed here. Nevertheless, these

simple scenarios provide a useful baseline for interrogating how spatially correlated error can be diagnosed from inter-scene

differences in *nMCC*. With respect to systematic error, I only considered the case where truth and model data are offset by one

pixel. Despite its simplicity, this exercise revealed that the scale of individual features matters when error is correlated to

feature location. Systematic error scenarios produced an asymmetrical sensitivity of *nMCC* to the fraction of bedrock, a result

that was not an artifact of ignoring components of the confusion matrix. This result begs the question as to what other properties

of features are changing as a function of bedrock fraction that can explain the observed asymmetry (Figs. 5-6), a question

which I explore in more depth below.

### 6.2 Size and shape of features

Up to now, the focus has been on what to expect from pixel-level accuracy scores when a binary classifier for bedrock is

applied across a gradient in bedrock fraction. Embedded in this analysis are assumptions for how outcrops emerge at higher

bedrock fractions. Specifically, I assumed that the spatial distribution of incipient features is random. This treatment allowed

me to probe how scene-level and pixel-level accuracy relate when sampling across large gradients in bedrock exposure. A

negative correlation between *nMCC* and bedrock fraction emerged in scenarios with systematic error, regardless of incipient

tor size (Fig. 5B; 6B). Given that *nMCC* addresses the problem of asymmetry with respect to target class (Fig. 3), what causes

this asymmetrical sensitivity of *nMCC* to systematic error?

In all the scenarios I have presented, the minimum tor size is set by the tor length. Because incipient tors are placed on the

surface by randomly placing their centres in the scene, more complex features are generated where incipient tors overlap by

chance. To illustrate the implications of this approach, Figure 7 examines how feature size and shape: 1. Impact the error

caused by a 1-pixel shift, and 2. Change as a function of bedrock fraction for the scenarios considered in this study. In general,

error is expected to go down with increasing feature size because area increases faster than the length of the edge of the feature

being offset. Due to the symmetry of the error induced by a feature being offset by one pixel, *recall*, *precision*, and *F1-score*

are equivalent for this kind of systematic error (Fig. 7A). I focus on these values as a measure of error induced by feature shape

alone that is independent of the scene-level bedrock fraction. As bedrock fractions increase in my synthetic surfaces, the

average size of individual features gets larger. Whether this increase in feature size is due to changing the incipient tor size or

the coalescing of many incipient tors into a single feature, *F1-scores* always monotonically improve (Fig. 7B). However,

Figure 7A nicely contrasts the differences in the error induced by a 1-pixel shift of simple features like square tors versus the

more complex ones generated by the coalescing of incipient tors. For the same feature area, simple feature boundaries produce

less error than sinuous, convexo-concave boundaries because the area to edge ratio is higher, thereby minimizing the impact

of translational offsets (see more examples in Fig. B1). Most shapes produce less error as they get larger, though it is possible





to create shapes that produce more error as they get larger (see 'star' shape in Fig. B1). In the synthetic scenarios where there is systematic error, both the average feature area and *F1-scores* increase with increasing bedrock fraction. The error looks like that of isolated square tors only at the lowest bedrock fractions. As features get larger, *F1-score* substantially improves but at

a lower rate than if bedrock was modelled as a single square tor (black line in Fig. 7B), reflecting the lower area to edge ratios produced by these complex feature shapes. Interpreting *F1-score* is limited by the fact that is does not account for True Negatives, which necessarily go down as bedrock fraction goes up. As such, increasing bedrock fraction in my synthetic scenarios should record the trade-offs between increasing feature sizes leading to less error and increasing feature complexity leading to more error. The negative trends in *nMCC* shown in Figures 5B and 6B suggest that the net result of these competing

effects is that increased complexity is the dominant term. The asymmetrical sensitivity of *nMCC* to systematic error also highlights the importance of how feature abundances are being simulated. Are the subsequent bedrock maps produced in this study representative of the actual transition from soil-mantled to bedrock-dominated hillsides? I cast this question more broadly in the section below where I can consider how the genesis and growth of features is embedded in pixel-level scores.

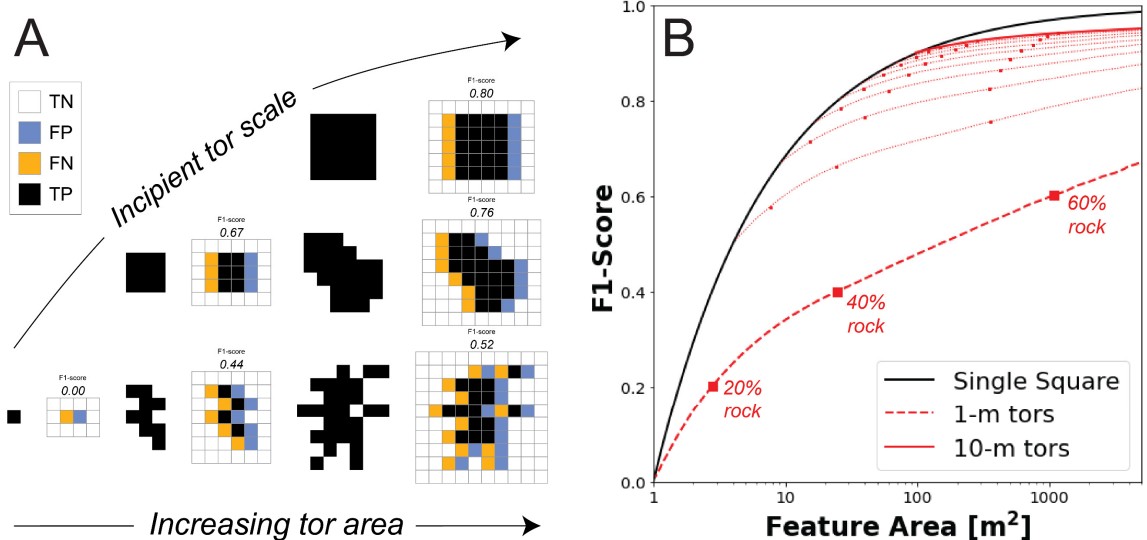

**Figure 7:** The frequency of False Positives and False Negatives (A) is a function of the incipient tor shape and average tor size, which can be quantified for (B) the systematic error scenario using *F1-score*. In A, six permissible tor shapes are shown for three different incipient tors sizes (rows) and three different tor areas (columns). The incipient tor shape both controls the minimum feature scale and the complexity of feature boundaries, whereby smaller incipient tors can produce more complex shapes and higher error rates for a given feature size (see associated *F1-scores*). In B, the *F1-score* is plotted as a function of 'average feature area for all the scenarios shown in Figure 5 (i.e.,

systematic error only). Square markers indicate values at three bedrock fractions. While average tor sizes get larger as more bedrock is exposed, this relationship is strongly contingent on the incipient tor size. Error at the lowest bedrock fractions largely reflects the error associated with imposed incipient tor shape and size. To illustrate this, the *F1-score* for square tors of a given area is also shown for reference (black line).



### 6.3 Other geomorphic applications

Many geomorphic tasks share the need for binary classifiers that perform well across gradients in feature abundance. Whether constraining the density of landslide scars, river channels, bedrock outcrops, or pit-mound features, geomorphic studies often rely on fine-scale mapping to determine how feature size, extent, and prevalence respond to differences in environmental forcing. As such, there is a general need for classifiers that successfully handle imbalanced data. In all the synthetic scenarios presented here, increased target class density was generated by randomly distributing the nuclei of incipient features within

the model domain. This is perhaps a reasonable analogue to the case where bedrock tors are exhumed from a spatially random distribution of somewhat more resistant bedrock (e.g., due to differences in chemical composition, fracture density, etc.) that underly thin soils near denudational thresholds. In contrast, many topographic features show striking evidence for self-organization (Hallet, 1990; Phillips, 1999; Murray et al., 2009) where feature properties instead reflect interactions of local positive feedbacks and far-field negative feedbacks (e.g., Gabet et al., 2014). Unlike the synthetic surfaces shown here, the

emergence of patterned topography and the regular spacing of features will maintain isolation of features even at very high densities. In such cases, we might expect the shape and size of features to follow well-defined scaling laws that respond quite differently to systematic error.

It is beyond the scope of this analysis to test the variety of scaling relationships that different topographic features exhibit in

nature. That said, this analysis emphasizes the importance of understanding how feature size and shape covary with each other as feature density increases. If scaling relationships do exist for a given type of feature and are known, then they provide the baseline for interpreting how and whether pixel-level accuracy scores are differentially sensitive to feature abundance. Even though pixel-level metrics like *MCC* and *nMCC* handle imbalanced data well and address the challenge of asymmetry with respect to target class, the examples shown here suggest that it should not be assumed that these pixel-level metrics will be

invariant as a function of feature abundance. How pixel-level metrics vary under different error scenarios need to be modelled explicitly so that trends in accuracy can be interpreted. The approach taken here was to use synthetic feature maps to yield insight into how pixel-level scores relate to scene-level attributes (sp., bedrock fraction). Future work would benefit from using landscape evolution models to inform how pixel-level scores are expected to vary under different error scenarios for the relevant geomorphic processes at play. As numerical models of the land surface attempt to keep pace with increasingly higher

resolution, process-scale observations (Tucker & Hancock, 2010), they have the potential to provide hypothesis-driven statistical analysis for how pixel-level accuracy scores should vary with feature abundance for different types of error.

In many cases, we expect error to depend on the topographic proxy being used (e.g., slope, curvature, roughness) such that error may be higher in scenes closer to the feature detection limit (i.e., where fewer features are observed). As such, more

careful consideration of spatial autocorrelation in error and the subsequent trends in accuracy scores that arise is needed. Further attention to this issue will undoubtedly reveal different relationships between pixel-level scores and scene-level




attributes than those presented here. Nevertheless, the error scenarios considered reveal that the domain over which *nMCC* is expected to be comparable across scenes can be quite limited depending on the source of error, the error rate, and the size and shape of features being assessed.

**6 Conclusions**

With increasing access to high resolution data and increasing focus on fine-scale mapping of topographic features, pixel-level accuracy assessment provides a powerful tool for understanding how well classifiers built from lidar topography are performing. To be most useful, the limitations of commonly used metrics like *precision*, *recall*, and *F1-score* need to be considered. Classification tasks that span large gradients in feature abundance are particularly vulnerable to biases in these

metrics because data is strongly imbalanced and the choice of target class matters. More robust metrics like *MCC* and *nMCC* largely address these methodological challenges. However, caution is still warranted in comparing pixel-level scores across gradients in feature density and extent (e.g., bedrock fraction). If error is random and uniform across scenes, then *nMCC* will dramatically worsen near endmember cases because the more prevalent class will be preferentially modified. If the model is systematically offset from the truth grid, then an asymmetrical sensitivity of *nMCC* can arise depending on the assumptions

for the genesis and growth of individual features. As the size of individual features increases with feature abundance there will also be lower sensitivity to systematic offset. However, if the shapes of features are also getting more complex, then the increased edge to area ratio of individual features can counteract and exceed improvements in accuracy associated with larger feature sizes. Though pixel-level metrics used in the machine learning and remote sensing community should be more widely adopted in geomorphic research, further work is needed to understand how different sources of error decouple pixel-level from

scene-level measures of accuracy.

**Appendix A: Random error and accuracy**

Section 5.1 reported how pixel-level accuracy scores vary as a function of bedrock fraction for a fixed rate of random error. While the synthetic surfaces were generated using Python, the results shown in Figure 4 can be directly derived from the mean random error rate ($\bar{e}_r$) and true bedrock fraction ($f_b$) analytically. Under this scenario, the probability of flipping either class

is independent of the prevalence and location of bedrock outcrops such we can define the average frequencies for all four components of the confusion matrix. The relative frequencies of each outcome are the product of the average rate of error (or non-error) and the average abundance of the true class. For example, the True Positives reflect both the probability of bedrock occurring ($f_b$) and the probability of not being flipped in the model due to random error (i.e., $1 - \bar{e}_r$). The frequencies of all four classification outcomes are:

$$f_{TP} = (1 - \bar{e}_r)f_b \tag{A1}$$
$$f_{FP} = \bar{e}_r f_b \tag{A2}$$



$$f_{FN} = \bar{e}_r(1 - f_b) \tag{A3}$$

$$f_{TN} = (1 - \bar{e}_r)(1 - f_b) \tag{A4}$$

Because we also know that the bedrock fraction in the model ($f_{bm}$) must equal the sum of the fractions of True Positives and False Negatives, these equations yield the relationship:

$$f_{bm} = (1 - \bar{e}_r)f_b + \bar{e}_r(1 - f_b) \tag{A5}$$

Equation A5 can be rearranged and simplified to describe how model bedrock fraction and true bedrock fraction vary as a linear function of the random error rate:


$$f_{bm} = (1 - 2\bar{e}_r)f_b + \bar{e}_r \tag{A6}$$

The relationships shown in Figure 4A (main text) are equivalent to equation A6 for different error rates. That the Python-generated scenes match the analytical solution indicates that the domain used for these synthetic scenes is large enough to adequately sample population statistics. Note that equation A6 provides a prediction for the relationship between true and model bedrock fractions only if error is uniform and random across scenes. In such cases, the average error rate can be directly

inferred from both the slope and y-intercept of the regression. If this reasoning is flipped, then empirical studies using scene-level regressions (DiBiase et al., 2012; Rossi et al., 2020) provide *prima facie* evidence for whether classification error is random and uniform across scenes. For example, while all regressions reported in Rossi et al. (2020) were forced through the origin, the best-fit linear regressions yielded negative y-intercepts suggesting that error rates were systematically higher at lower bedrock fractions.


Because pixel-level accuracy scores can be derived directly from the confusion matrix, the simplified assumptions of random, uniform error also facilitate prediction for how *F1-score* and *nMCC* will vary with the true bedrock fraction. Substituting the values from eqs. A1-A4 into equation 5 (main text) yields:

$$F1\text{-}score = \frac{2f_b(1 - \bar{e}_r)}{2f_b(1 - \bar{e}_r) + \bar{e}_r} \tag{A7}$$

which is equivalent to the numerically generated black curves in Figure 4B. Similarly, substituting eqs. A1-A4 into equation 6 (main text) yields:

$$MCC = \frac{\sqrt{f_b} \times \sqrt{1 - f_b} \times (1 - 2\bar{e}_r)}{\sqrt{f_b + \bar{e}_r - \bar{e}_r^2 - f_b^2 - 4\bar{e}_r f_b - 4\bar{e}_r f_b^2 - 4\bar{e}_r^2 f_b - 4\bar{e}_r^2 f_b^2}} \tag{A8}$$

which is equivalent to the numerically generated red curves in Figure 4B. Though the expression for *MCC* under random, uniform error is complex, it reveals why there is strong and symmetrical sensitivity near the endmember cases of zero and all

bedrock. The numerator in eq. A8 decreases faster than the denominator near endmember cases regardless of the average error rate. Since $f_b$ and $1 - f_b$ are complementary and $\bar{e}_r$ is assumed to be constant, this reduction in *MCC* is also symmetrical around an optimal bedrock fraction of 0.5.



## Appendix B: Feature shape and systematic error

In this analysis, bedrock tors are treated as square features whose scale is varied with a single parameter, the 'tor' length. The
square geometry is useful because it is oriented in the same way as the regular grid over which the synthetic landscapes are generated. The random placement of incipient tors on the surface ensures that bedrock features do not have a preferential orientation and translational errors do not depend on the orientation of offset. While relaxing these assumptions are beyond the scope of this study, it is worth probing more deeply on how tors are simulated to help explain the asymmetrical sensitivity of $nMCC$ to bedrock fraction when the model data is systematically offset from truth (Figs. 5-6). Specifically, I show in this
appendix how the frequency of False Positives and False Negatives are linked to the shape and size of features. Both the incipient tor shape and the subsequent aggregation of these shapes into larger bedrock features are what set the overall error rate. By incipient tor shape, I am referring to the seed shape used to generate bedrock from the random placement of tor centres. While I only used a square seed in the main analysis, Figure B1 shows the importance of seed shape to generating false positives and false negatives when the truth and model features are systematically offset by one pixel. In Figure B1A, I show
four shapes at three different spatial scales. Because seed shapes are constrained by their raster representation, it is hard to create different shapes with the same area when the seed shape is small. For the shapes 'square', 'rounded', 'plus', and 'star', the shape area is approximately equivalent for shape diameters of 6, 7, 10, and 11 pixels, respectively. For these radially symmetrical shapes, a 1-pixel shift produces the same number of false positives and false negatives regardless of the orientation of the shift.



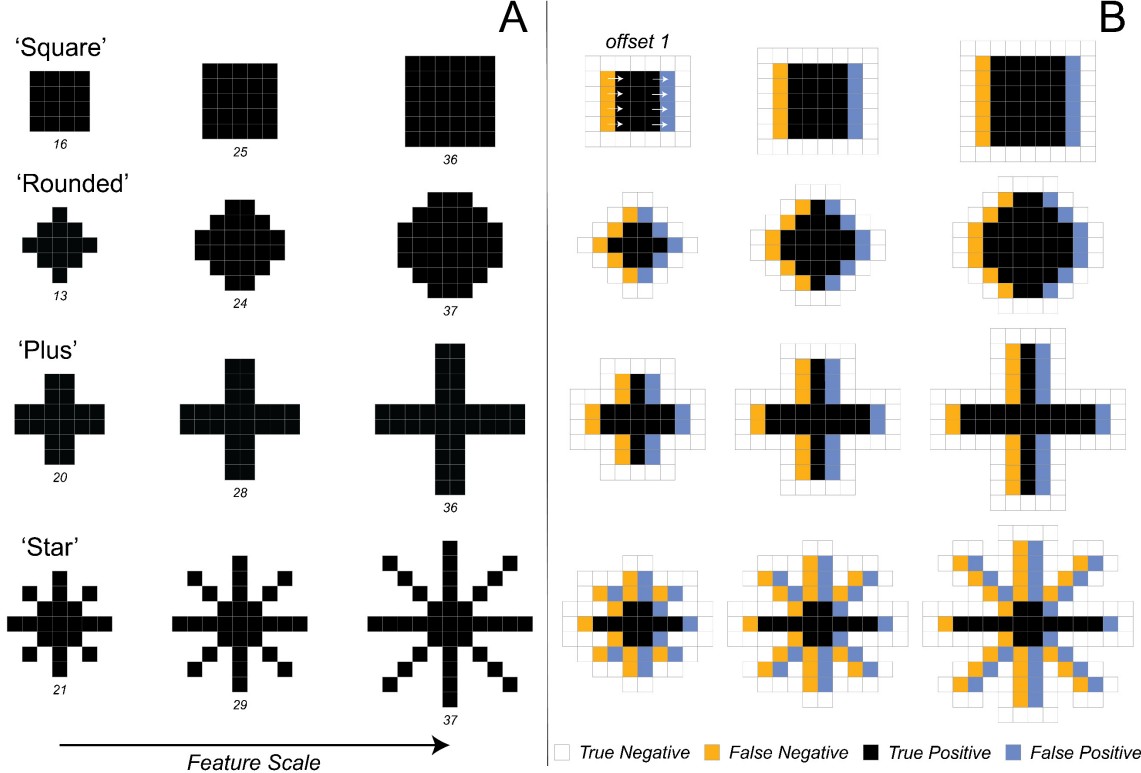

**Figure B1:** (A) Shape and scale of incipient tors directly affects (B) the subsequent frequencies of False Negatives (yellow) and False Positives (blue) to a translational offset in model classification. In A, four different feature shapes are shown that have either convex (i.e., square, rounded) or concavo-convex (i.e., plus, star) boundaries with respect to the soil matrix. The feature area is reported below each shape in pixels. Note that the smallest 'rounded' example is not actually round, but a rotated square. As features get too small with respect to the data resolution, it becomes difficult to represent complex objects using a regular, square grid. In B, error classes are shown for a 1-pixel shift to the right. Because shapes are all rotationally symmetric with respect to the four cardinal directions, error rates do not depend on the direction of the shift. Only true negatives that share an edge with the other classes are shown.

While much of the analysis has emphasized that *MCC* and *nMCC* are superior to *F1-score* for accuracy assessment when True Negatives matter, *F1-score* is well-suited to the task of isolating how feature size and shape impact error independent of bedrock fraction. The geometry of an individual square tor is a useful starting point because the error induced by a one-pixel shift between truth and model classification is readily derived from its simple geometry. The number of True Positives is equal to $l^2 - l$ and the number of False Positives and False Negatives are each equal to $l$, where $l$ is the length of the square. Substituting these terms into eq. 5 (main text) and simplifying yields an equation for *F1-score* specific to square features:

$$F1\text{-}score_{sq} = 1 - \frac{l}{l^2} \tag{B1}$$





Because the area of a square increases faster than its length, the last term in equation B1 explains why accuracy improves as a function of feature area. To extend this further, I also compare how the different shapes shown in Figure B1 impact *F1-scores* (Fig. B2B). The key property that is changing for these different shapes is the edge to area ratio, whereby concave shapes (i.e., 'square' and 'rounded') are generally more conducive to higher *F1-scores*. However, the size of features is also very important. In all but the 'star' example, *F1-scores* improve as features get larger. The 'star' does not follow the pattern because it is the

only example here where the edge to area ratio increases with feature size.

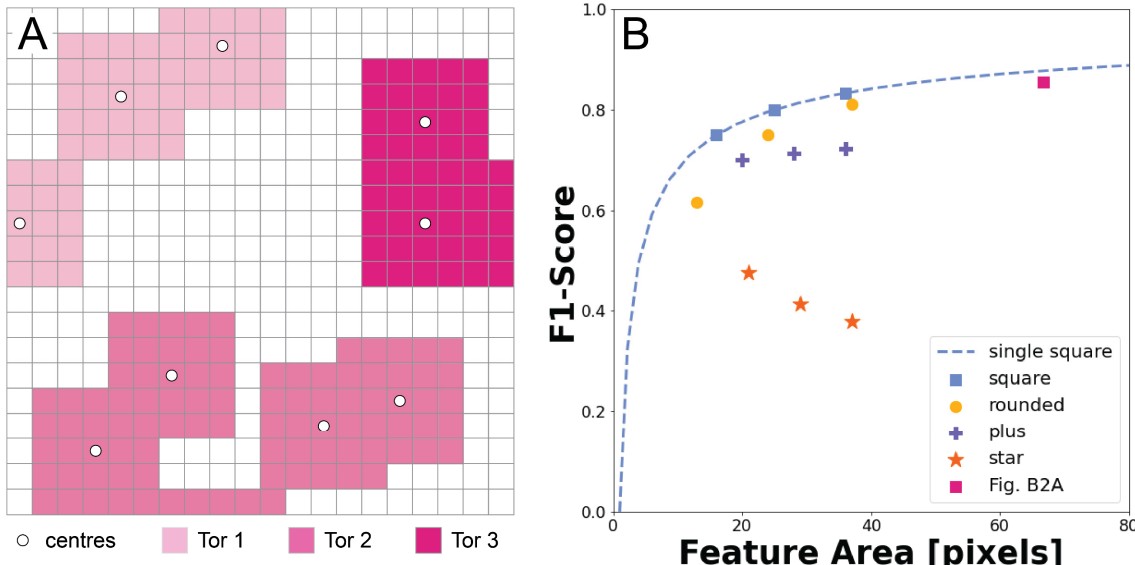

**Figure B2:** (A) Example bedrock map like those produced in this analysis and (B) the relationship between feature area and *F1-score*. In A, nine 'tor' centres produce three individual tors. In B, the *F1-score* in A is plotted as a function of the average feature area alongside the seed shapes shown in Figure B1. For reference, I also plot the function describing how area and *F1-score* vary for square tors. Note that even 610 though the bedrock map shown in A is generated using randomly distributed square outcrops, the *F1-score* is lower than the dashed blue line due to the more complex boundaries generated by coalescing incipient tors into a fewer number of features.

This feature level assessment provides further insight into why Figures 5-6 (main text) show positive trends in *F1-score* and negative trends in *nMCC* as a function of bedrock fraction. Increasing feature size favours lower error while increasing feature complexity favours higher error. Figure 7 (main text) demonstrates that the net result of these competing effects is monotonic 615 increases in *F1-score* as a function of bedrock fraction, all other things being equal. This is because it is difficult to overcome the strong sensitivity of *F1-score* to how balanced the data are. Instead, *nMCC* reveals that the impact of increasing complexity of feature shape slightly outweighs the increase in feature size leading to a modest negative relationship to bedrock fraction (Fig. 5 main text).



**Data Availability**

Figures 1-2 and Table 1 are based on the bedrock mapping at site P1 from Rossi et al. (2020). Maps for 1-m truth and model data at this site can be accessed at https://github.com/mwrossi/cfr_extremes. These classified maps are based on 2018 Pictometry® orthomosaicked air photos purchased by Boulder County and airborne lidar data acquired by the National Center for Airborne Laser Mapping for the Boulder Creek Critical Zone Observatory (Anderson et al., 2012). Synthetic surfaces presented in Figures 3-7 were built in Python. Scripts can be accessed at https://github.com/mwrossi/bedrock-mapping-

accuracy. Once through review, the main code will continue to be hosted on Github, but scripts and files used for generating figures will be archived on Figshare.

**Competing interests**

The author declares that there is no conflict of interest.

**Acknowledgements**

This project was supported by funding from the Geomorphology and Land-Use Dynamics Program at the National Science Foundation (EAR-1822062) and benefited from discussions with Bob Anderson, Suzanne Anderson, Roman DiBiase, Brittany Selander, and Greg Tucker.

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
