# Peer review of "Short Communication: Evaluating the accuracy of binary classifiers for geomorphic applications"

_Earth Surface Dynamics, 2022_

## Author Response (AR1)

Dear Editors,

Both reviewers were positive in their review of '*Short Communication: Evaluating the accuracy of binary classifiers for geomorphic applications*' and provided very helpful feedback that has improved the quality of the manuscript. First off, I apologize for the delay in submitting revisions to this manuscript. While the feedback from both reviewers was that no new analysis was needed, the new figures and reframing of the discussion required a few iterations before I felt the new version of the manuscript was ready.

In my view, the reviewers homed in on four major points: (1) The analysis is too long for a Short Communication; (2) The bedrock mapping example obscures the more general findings on binary classification; (3) The shape analysis from former Appendix B needs to be elevated to the main text, and (4) There is a need for a clear statement of best-practices for using pixel-level accuracy metrics.

In response to 1, the manuscript has gotten (slightly) longer during revisions, especially with the addition of two new figures. Given that both reviewers felt it was too long in the previous round, I request to change the manuscript type from a 'Short Communication' to a standard 'Research Article.'

In response to 2, Reviewer 1 (Grieve) provided two suggestions—either cast the whole manuscript more generally or re-organize the manuscript and restrict the bedrock mapping example as a case study. Reviewer 2 also found the back and forth between binary classification, in general, and the bedrock mapping application, in particular, hard to follow. As such, I decided to re-frame this manuscript as a more general treatment of binary classifiers. All figures now refer to features and feature objects instead of bedrock exposure and tors; the old section 2 titled 'Example application: Bedrock mapping' has been removed; and the old Figure 1 focusing on bedrock mapping along Boulder Creek has been replaced with a more generic figure showing three exemplar geomorphic applications that use binary classification. While this re-framing took a bit of effort, I agree with the reviewer sentiments that this analysis is stronger as a general contribution that is not bogged down in the specific issues associated with the transition from soil-mantled to bedrock-dominated hillsides.

In response to 3, I agree with the reviewers that the analysis and discussion of feature shape and scale is exciting. Taking their suggestions as my lead, I have now integrated the discussion and figures from the old Appendix B into the main text (mostly in the new section 5.2.1).

In response to 4, I have now re-written section 5.3 to be a 'Recommendations and Future Directions' section with a numbered list of recommendations and a numbered list of promising future research directions.

I think you and the reviewers will appreciate the revised manuscript, and I look forward to it being published in Earth Surface Dynamics, in due course. My detailed responses to reviewer comments are provided as in-line responses below (my responses in blue).

Cheers,
Matt

**Reviewer 1: Stuart Grieve**

I'd like thank the author for this contribution, which feels particularly relevant as machine learning approaches become increasingly common in geomorphology research. This manuscript explores the evaluation of the accuracy of binary classification of raster datasets, using mapping of bedrock tors as a case study. There is some discussion and presentation of the methods used in classifying real data, distinguishing between bedrock and soil within forest canopy gaps. A series of synthetic datasets are generated with known error properties and bedrock to soil ratios. These synthetic datasets are used to identify trends in two accuracy metrics, F1 Score and Matthews Correlation Coefficient, at a range of bedrock fractions when errors are random, systematic or both. A key result is the elegant demonstration of the unsuitability of using F1 Score as an accuracy metric in most settings, with the alternative Matthews

Correlation Coefficient performing more robustly. There is then analysis of changes in bedrock tor shape and size with systematic error, demonstrating the sensitivity of these metrics to shape and size variation.

I believe that this Short Communication fits well into the Earth Surface Dynamics remit, and will be a valuable addition to the broader computational geomorphology literature. Overall, I am in favour of this manuscript being published, and look forward to seeing a final version in due course.

Thank you for your constructive and supportive feedback!

General comments

Overall, this is a well written and presented manuscript which is methodologically and theoretically sound. I do not have recommendations or requests for additional analysis, but have some observations on the presentation and structure of the manuscript that I hope the author will find useful.

Taken as a whole, for a short communication, the manuscript feels quite long and in places dense. One of the things that I struggled with when reading this manuscript, was whether it mattered that this was a manuscript about bedrock tors and the emergence of bedrock. Indeed, the paper's title does not mention bedrock at all. I wonder if it would streamline the paper to remove much of the discussion of bedrock tors, in favour of speaking more generically about binary classification of geomorphic data. I realise there is a fine line to tread here to keep the geomorphic relevance and novelty of the work, but I feel that there are a lot of tangential issues that could be raised around this paper regarding the correct mapping of tors, fuzzy boundaries, mixed pixels, etc, none of which are relevant to the evaluation of binary classifications. Particularly when approaching the impact that feature shape has on classification and error - this is really cool and could have broad implications for a range of applications but risks being bogged down in discussion of bedrock structure and emergence, rather than the more theoretical advancements being focused on.

I agree with the reviewer's sentiment that this manuscript may be better served as a more general analysis instead of getting bogged down in the specifics of classifying bedrock. This reviewer offers two solutions: (1) Keep the whole thing general, or (2) Reorganize the manuscript so that the bedrock mapping case study is introduced at the end (see next comment). I opted for the former because I was convinced by both reviewers that the bedrock discussion made it hard to follow the main analysis, which is feature agnostic. To this end, there were a number of revisions made. First, the old Figure 1 focusing on bedrock mapping has been replaced with a three-panel figure showing different kinds of features well-suited to binary classification (mima mounds, bedrock exposure, gully erosion). I retained Figure 2 from the bedrock mapping analysis of Rossi et al. (2020), though it is only referenced as an illustration for how F1-score and MCC are calculated from the confusion matrix. Second, the old section 2 ('Example application: Bedrock mapping) has now been removed. Third, all figures and text have been updated the terms 'bedrock fraction' and 'tors' with 'feature fraction' and 'feature objects,' respectively. Fourth, the Discussion section (new section 5) has been dramatically re-written in response to other reviewer comments. As such, much of the residual discussion on bedrock mapping was removed.

A potentially less extreme restructure of the manuscript could be to present all of the synthetic landscapes first, without any reference to bedrock, simply setting the work up as an evaluation of binary classification metrics. Then the final section of the manuscript can bring in the bedrock mapping as a case study, thereby tying the synthetic data and theoretical work to a geomorphic context, without introducing as much ambiguity and complexity into the earlier sections of the manuscript.

See response above. I decided to attempt the more drastic restructuring of the manuscript because I think it will make for a clearer and stronger manuscript. I hope the reviewers agree.

When presenting the results of the comparison between MCC and F1 scores, in the case of asymmetric metrics (eg Table 2), more could be made of this result. This may be a well known issue in the data science literature, but I think it can be highlighted in a disciplinary context here, and the implications of (mis-)use of F1 scores explored in a geomorphic context.

The reviewer is correct that the challenge of asymmetry is a well-known issue with F-measures in the data science literature, though I also agree that most readers will be unfamiliar with this behavior of F1-sore. I have largely retained the same discussion of asymmetry in the introduction to the accuracy metrics (lines 151-156; Table 1). However, I have now added a more in-depth discussion of asymmetry (lines 371-378) to help introduce the revised discussion on imbalanced mapping tasks in geomorphology (section 5.1). This includes a new figure (Figure 7) that compares nMCC to a modified version of F1-score that is symmetrical. Discussion of this figure (lines 387-401) argues that nMCC should still be favoured over the symmetrical calculation of F1-score because it is less sensitive to the random error case.

The work on tor shape in Section 6.2 and Appendix B is very exciting. I would prefer to see it all within the main manuscript rather than split out into an Appendix. There are a lot of interesting factors at play here, where binary classifications have to contend with systematic errors, data resolution and feature size. As data resolution improves, features such as bedrock tors are represented by more pixels, creating the potential for more complex shapes with increased edge to area ratios, and therefore these results are critical in beginning our understanding of how to best classify and evaluate those classifications.

I agree that the analysis on size and shape is quite exciting and was perhaps unsure in the original submission others would agree. Given that both reviewers liked this portion of the analysis, I have now revised the discussion section on the size and shape of features (section 5.2) to include much of the discussion and figures from the old Appendix B. Specifically, section 5.2.1 is what used to be in Appendix B and is now used to introduce the analysis of the features that emerged from synthetic scenarios (5.2.2). The two figures from the old Appendix B were combined into a new Figure 8. Finally, the point about increasing data resolution is well taken, and it is now highlighted as one of the 'promising future research directions' (lines 582-586).

A final recommendation for this manuscript would be the addition of a clear section of recommendations for the types of evaluation method to use under different generalised circumstances. I think this would help with the impact of the manuscript, by giving readers clear direction which they can feed back into their own work.

This is a good idea that both reviewers suggested. The final discussion section (section 5.3) is now titled 'Recommendation and future directions.' This dramatically revised section includes numbered lists of 'recommendations' (lines 543-558) and promising 'future research directions' (lines 569-586) that should help provide clarity on what was learned from my analysis and which important open questions remain.

Line by line comments

In addition to the comments above, I have some more general minor line by line comments:

Line 54 - Missing 'as' between 'long their'.

Fixed.

Line 108-110 - This sentence needs a citation.

This whole discussion of error was eliminated when I removed the 'Example application: Bedrock mapping' section.

Code

It is great to see the code associated with this manuscript available online, with the author detailing how they will archive it once the manuscript is accepted. I have gone through the code on github and it is well written and structured, and after some brief testing it appears to do what is described in the manuscript. It would be ideal if the repository had a licence file included, to ensure that people can use the code in the future. If you need help with software licences, you can look at https://choosealicense.com/ to guide you

through the process. To aid reproducibility it would also be helpful to record the numpy, scipy, matplotlib and python versions you are using within the readme, in case future upgrades break things in your code.

The code and github repo have now been updated to remove reference to 'bedrock' in the repo and filenames. Versions of the packages are now provided in the README file and docstrings of the scripts. A licensing file (MIT) is now added to the github repo as well.

-- Stuart Grieve

Reviewer 2: Anonymous

Review of the manuscript "Short Communication: Evaluating the accuracy of binary classifiers for geomorphic applications" from Rossi Submitted to Earth Surface Dynamics.

I would like to know the author for this very nice paper. This paper attempts to evaluate the accuracy of binary classifiers commonly used in raster datasets, in this case applied to the mapping of rock formations. In this study, the synthetic datasets (with known errors) were used to test two accuracy metrics F1 score and Matthews correlation coefficient on a range of bedrock fractions when errors are random, systematic and both, as well as variable shape and size, showing that Matthews correlation performs more robustly.

Thank you for your constructive and supportive feedback!

I think this short communication fits well into the Earth Surface Dynamics journal. However, I believe that this type of work would fit better in a broader computational journal (e.g. IEEE) as it makes a valuable contribution as machine learning approaches become increasingly common in geomorphological research.

While I generally agree that this manuscript could fit nicely into a broader computational journal, I hope it will be continue to be considered for publication in Earth Surface Dynamics for a couple of reasons. First, it is well within the scope of the journal. Second, and more importantly, my target audience is geomorphologists who are using lidar data for feature classification, but who may or may not be familiar with these methods. While the metrics being presented here are widely known and used in the remote sensing and machine learning communities, they have only been intermittently adopted in the broader geomorphological community. Nevertheless, I also agree that some of the insights offered are relevant to ongoing research in image segmentation and machine learning. I am hopeful it will still be seen by these communities even if it is published in a more disciplinary journal.

General comments:

The paper is well written and presented. I don't have any further recommendations or additional analysis in the methodology and results section. However, I have very few concerns, mainly related to the way this paper is presented.

The title refers to a "short communication". However, the text is quite long. There are some places where it is not easy to read, mainly because the focus is on the binary classifiers (which could be very ambiguous) and very little on the geomophical implications (i.e. bedrock). I think this paper would be better suited as a research paper rather than a short communication. I would suggest reorganizing the different sections, showing the methodology first and then presenting the geomorphological applications as a study case.

I agree that this manuscript has expanded beyond a 'Short Communication.' Both reviewers found this paper too long and too dense to qualify as such, and I agree. I have requested to have this paper submitted as a regular Research Article. In response to Reviewer 1, I reframed the whole analysis more generally. The geomorphic implications are now also more general and not focused on the specific

application of bedrock mapping. Hopefully, this will aid in the readability of the manuscript. The paper no longer jumps back and forth between binary classification, in general, and bedrock mapping, in particular. Finally, the new Figure 1, revised introduction, and revised discussion now try to contextualize this analysis using a broad suite of potential geomorphic applications. Hopefully, these additions help tether the more abstract analysis of binary classification to different real-word scenarios. More specifically, I've highlighted three use-cases that are decidedly different from how features are treated in this analysis— namely mima mounds that have a characteristic scale and spacing, bedrock exposures whose properties can be very heterogeneous, and gully erosion which is directional and isotropic. These examples are used as the basis for recommending how this analysis can be extended in future studies (see lines 563-586).

I suggest changing the verb conjugation in some sentences, especially where it says 'I did' or 'I presented', to more neutral third person verb tenses (e.g. This study, on this work, etc).

I went through the manuscript and tried to remove these kinds of phrases. It did lead to more frequent use of passive voice though.

Appendix B should appear in the discussion section. This is a valuable contribution to the newer remote sensing products (e.g. higher resolution) from which the patterns presented in Figure B1 originate.

*Copied from response to first reviewer*:
I agree that the analysis on size and shape is quite exciting and was perhaps unsure in the original submission others would agree. Given that both reviewers liked this portion of the analysis, I have now revised the discussion section on the size and shape of features (section 5.2) to include much of the discussion and figures from the old Appendix B. Specifically, section 5.2.1 is what used to be in Appendix B and is now used to introduce the analysis of the features that emerged from synthetic scenarios (5.2.2). The two figures from the old Appendix B were combined into a new Figure 8. Finally, the point about increasing data resolution is well taken, and it is now highlighted as one of the 'promising future research directions' (lines 582-586).

Finally, I suggest adding a recommendation section regarding the perspectives of this very nice work, especially regarding the limitations.

Thank you.

*Copied from response to first reviewer*:
This is a good idea that both reviewers suggested. The final discussion section (section 5.3) is now titled 'Recommendation and future directions.' This dramatically revised section includes numbered lists of 'recommendations' (lines 543-558) and promising 'future research directions' (lines 569-586) that should help provide clarity on what was learned from my analysis and which important open questions remain.

---

## Author Response (AR2)

Dear Editors,

I made minor grammatical edits and fixed the two technical corrections identified by Reviewer Stuart Grieve.

I am not seeing a place where I can change the manuscript type from Short Communication to a regular Research Article. Are you able to make this fix on your end. Thanks.

Cheers,
Matt

**Reviewer 1: Stuart Grieve**

I'd like thank the author for their diligent redrafting of this manuscript following comments from myself and another reviewer. In general I am really pleased with the new, more focused, manuscript which has a clear message about the use of binary classifiers within geomorphology. A major issue I had with the original version of this manuscript was in the mixed focus between bedrock emergence and binary classification. The author has now addressed this, and the result is a much stronger, more broadly relevant contribution. The expansion of the work considering systematic error and feature shape is an excellent addition to the manuscript and presents an interesting avenue for future research in this area.

Going through the manuscript and the author's response to review comments, I am happy that this revision addresses all of my prior concerns and I look forward to seeing this manuscript published in due course.

I spotted a couple of minor issues that should be sorted prior to publication:

Figure 1C's title reads "Sant Cruz" rather than "Santa Cruz".

Fixed.

Figure 2B's confusion matrix is difficult to read as one of the sections is a white text on a yellow background.

White text is now outlined in black to help with yellow square.